# Monitoring of Fabric Integrity and Attrition Rate of Dual-Active Ingredient Long-Lasting Insecticidal Nets in Tanzania: A Prospective Cohort Study Nested in a Cluster Randomized Controlled Trial

**DOI:** 10.3390/insects15020108

**Published:** 2024-02-04

**Authors:** Jackline Martin, Eliud Lukole, Louisa A. Messenger, Tatu Aziz, Elizabeth Mallya, Edmond Bernard, Nancy S. Matowo, Jacklin F. Mosha, Mark Rowland, Franklin W. Mosha, Alphaxard Manjurano, Natacha Protopopoff

**Affiliations:** 1Department of Parasitology, Pan-African Malaria Vector Research Consortium, Kilimanjaro Christian Medical University College, Moshi P.O. Box 2240, Tanzania; tatuaziz1992@gmail.com (T.A.); mallyaelizabeth@gmail.com (E.M.); fwmosha@gmail.com (F.W.M.); 2Department of Parasitology, National Institute for Medical Research, Mwanza P.O. Box 1462, Tanzania; ellylufi@ymail.com (E.L.); edmondbernardhakiz444@gmail.com (E.B.); jfmosha@yahoo.com (J.F.M.); amanjurano@gmail.com (A.M.); 3Department of Disease Control, Faculty of Tropical Diseases, London School of Hygiene and Tropical Medicine, London WC1E 7HT, UK; louisa.messenger@unlv.edu (L.A.M.); nancy.matowo@lshtm.ac.uk (N.S.M.); mark.rowland@lshtm.ac.uk (M.R.); natacha.protopopoff@lshtm.ac.uk (N.P.); 4Department of Environmental and Occupational Health, School of Public Health, University of Nevada, Las Vegas, NV 89119, USA; 5Parasitology and Vector Biology Laboratory (UNLV PARAVEC Lab), School of Public Health, University of Nevada, Las Vegas, NV 89119, USA

**Keywords:** long-lasting insecticidal net, median function survival, survivorship, attrition, fabric integrity, Tanzania

## Abstract

**Simple Summary:**

This study evaluated the physical durability of new types of bed nets with two insecticides incorporated into the fibers (Interceptor G2, Royal Guard, and Olyset Plus) compared to standard nets (Interceptor), which contain a single insecticide (pyrethroid only). These bed nets were distributed in the Misungwi district, Tanzania, in February 2019 and followed up at 6-month intervals up to 36 months post-distribution. During cross-sectional surveys, householders were asked to use the net until the next survey. These nets were supposed to protect the user for three years, but this was not the case in this study. All net types had a life span of much less than three years including the pyrethroid-only net. In response to the questionnaire, most respondents reported that they discarded their nets due to wear and tear and this was evident from the holes accrued in earlier surveys; this effect was more severe with Olyset Plus nets than with standard Interceptor nets and other dual insecticide nets.

**Abstract:**

Pyrethroid-treated long-lasting insecticidal nets (LLINs) have been the main contributor to the reduction in malaria in the past two decades in sub-Saharan Africa. The development of pyrethroid insecticide resistance threatens the future of LLINs, especially when nets become holed and pyrethroid decays. In this study, three new classes of dual-active ingredient (AI) LLINs were evaluated for their physical durability: (1) Royal Guard, combining pyriproxyfen, which disrupts female fertility, and a pyrethroid, alpha-cypermethrin; (2) Interceptor G2, which combines the pyrrole chlorfenapyr and a pyrethroid (alpha-cypermethrin); (3) Olyset Plus, which incorporates the pyrethroid permethrin and the synergist piperonyl butoxide, to enhance the pyrethroid potency; and Interceptor, a reference net that contains alpha-cypermethrin as the sole active ingredient. About 40,000 nets of each type were distributed in February 2019 to different villages in Misungwi. A total of 3072 LLINs were followed up every 6–12 months up to 36 months to assess survivorship and fabric integrity. The median functional survival was less than three years with Interceptor, Interceptor G2, and Royal Guard showing 1.9 years each and Olyset Plus showing 0.9 years. After 36 months, 90% of Olyset Plus and Royal Guard and 87% of Interceptor G2 were no longer in use (discarded) due to wear and tear, compared to 79% for Interceptor. All dual-AI LLINs exhibited poor textile durability, with Olyset Plus being the worst.

## 1. Introduction

Pyrethroid-only insecticide-treated nets (ITNs) and long-lasting insecticidal nets (LLINs) were the cornerstone for malaria vector control until comparatively recently when pyrethroid resistance emerged and now threatens malaria control. Apart from resistance, other factors threaten the future of LLINs, including net fabric durability, insecticide efficacy and retention, net usage, and wear and tear by daily handling [1]. In areas with intense pyrethroid resistance, if LLINs are damaged, mosquitoes may penetrate the net holes and feed on human hosts, potentially transmitting malaria.

Formerly, the presence of LLINs with intact fabric (i.e., undamaged) provided a physical barrier that prevented human-vector contact and reduced human blood-feeding [2,3]; treating the nets with pyrethroid provided additional protection by adding a toxic, repellent barrier [1].

When mosquito populations become resistant and pyrethroid-only nets develop holes, users may perceive them as unprotective and discard them, leading to a reduction in coverage and usage [4,5].

Other studies have reported that when insecticide in the netting material decreases and nets acquire holes, users have no or minimal protection as the mosquitoes can penetrate and ingest blood [6,7]. A study conducted in Zambia showed that the poor fabric integrity of standard pyrethroid nets affected their effectiveness against *Anopheles arabiensis* [6], while another in Tanzania demonstrated that increased hole area was associated with higher numbers of *An. gambiae* inside the net [7].

Washing and drying LLINs have been reported to be among the factors that contribute to reduced LLIN insecticide concentration and the development of holes in the community [8]. Generally, social and economic status are two of the factors affecting net handling. A study conducted in Bouaké, Côte d’Ivoire, found that household owners with primary/higher education had better knowledge about how to manage (tuck in on the bed, washing, drying) nets than those who reported having received limited health information and education [9].

New classes of ITNs have been recommended by the WHO recently as they showed superior protection against malaria compared to standard LLINs in various cluster-randomized controlled trials (cRCTs) in Tanzania [10,11], Benin [12], and Uganda [13]. ITNs combining the synergists piperonyl butoxide (PBO) and pyrethroid have been recommended and deployed routinely since 2018. In 2023, two other ITNs, combining two insecticides, dual-active ingredient (AI), pyrethroid and either chlorfenapyr in Interceptor G2 or pyriproxyfen in Royal Guard, received WHO approval [14]. As these nets are being scaled up, net durability including fabric integrity and survivorship (attrition) [15] should be assessed to understand the epidemiological outcomes and how these interventions should be incorporated into vector control programs. As part of the cRCT in Tanzania, this study assessed the survivorship/attrition rate and fabric integrity of cohorts of three dual-AI ITNs (Royal Guard, Olyset Plus, and Interceptor G2) over 3 years of community use, compared to pyrethroid-only ITNs.

## 2. Methodology

### 2.1. Characteristics of the Long-Lasting Insecticidal Nets (LLINs) Tested

The current investigation was embedded within a comprehensive cluster-randomized controlled trial (cRCT) carried out in the Misungwi district, Tanzania [11]. In this cRCT, 84 clusters (21 clusters per intervention arm) received the distribution of four distinct types of long-lasting insecticidal nets (LLINs) in February 2019. The LLINs subjected to evaluation were as follows: (1) Royal Guard^®^ (Disease Control Technologies, LLC, Greer, SC, USA), a dual-AI LLIN comprised of polyethylene containing alpha-cypermethrin (261 mg/m^2^) and pyriproxyfen (225 mg/m^2^) known for its capability to disrupt female reproduction and fertility of eggs; (2) Interceptor^®^ G2 (BASF Corporation, Ludwigshafen, Germany)), a dual-insecticide LLIN made of polyester coated with wash-resistant formulations of chlorfenapyr (200 mg/m^2^) and pyrethroid (alpha-cypermethrin) (100 mg/m^2^); (3) Olyset^TM^ Plus (Sumitomo Chemicals, Tokyo, Japan), a LLIN that incorporates the pyrethroid permethrin (800 mg/m^2^) and the synergist piperonyl butoxide (400 mg/m^2^), which enhances the potency of permethrin; (4) Interceptor^®^ (BASF Corporation, Ansan, Republic of Korea), an alpha-cypermethrin-treated LLIN at a target dose of 200 mg/m^2^ coated onto polyester filaments as the reference intervention.

### 2.2. Study Area

Misungwi district covers an area of 2579 km^2^. The estimated total population in the area is 467,867 found in 78 villages. Notably, there has been a consistent 2.9% annual population growth observed from 2012 to 2022 [16]. The previous malaria control intervention in the area was a standard LLIN mass campaign conducted in 2015, indoor residual spraying (IRS) using pirimiphos-methyl from 2013 to 2017, and larviciding using Bti in 2018. The major malaria vector species found in the area are *An. funestus* complex, *An. gambiae* sensu stricto, and *An. arabiensis.* Details of the Misungwi cluster-randomized controlled trial (cRCT) have been previously published [17,18], providing comprehensive information on households and the number of nets distributed per arm. For the current study, a subsample of 20 study clusters out of the total 84 utilized in the cRCT were randomly chosen for the assessment of LLIN attrition and fabric integrity (see Figure 1). The complete protocol has been previously documented [17].

### 2.3. Study Design

This study adopted a prospective cohort design, tracking nets over three consecutive years to evaluate the survivorship/attrition and fabric integrity of potential dual-active ingredient (dual-AI) LLINs in comparison to standard LLINs. After LLIN distribution, a census/enumeration of households in the hamlet was completed as part of the cRCT, and each household was given unique identification numbers. Selected study LLINs were recorded and labeled with a household number and net number one month post-distribution.

### 2.4. Sample Size and Sampling

Sample size calculations were conducted using the power log-rank command in Stata v.15.1. A total of 750 LLINs per net type from 5 clusters per arm (equivalent to 150 per cluster) allowed for a detection rate with a 9.4% absolute difference (hazard ratio = 0.8651) in LLIN attrition rates, assuming an attrition rate in the control of 70% over the 3 years. This calculation takes into account an intra-cluster correlation coefficient (ICC) of 0.05.

Following distribution, all selected nets were labeled with the household number and a net number to generate a master list. In each arm, up to three nets from each selected household (HH) (with a total of 250 HHs selected) were assessed in 5 clusters per arm (20 clusters in total). The study nets (750 per arm) were randomly sampled from the master list and evaluated for survivorship/attrition and fabric integrity at 6, 12, 24, 30, and 36 months post-distribution. The objective of the study was explained to head of the household before net inspections and those who agreed to participate in the study were interviewed about their socioeconomic status, housing materials, and the condition of the net through a structured questionnaire and templates for hole assessment.

### 2.5. Attrition Rate

In this study, attrition rate was defined as the number of nets that were not present in the household due to wear and tear or other causes [19]. The reverse of the attrition rate was survivorship, which included all nets present in the household during the survey. All causes of attrition were assessed using a structured questionnaire. A structured questionnaire was employed during the survey and questions were asked in Swahili or the regional language depending on the preference of the participants. The physical presence of the nets was observed by field technicians. Probing questions were utilized to inquire about the net’s location, enabling owners to specify whether the net was discarded, given away, or used in another location. The procedure adhered to the WHO guidelines for the laboratory and field assessment of LLINs in 2013.

Differences in attrition rate were assessed as per WHO guidelines [20]. The attrition rate was assessed in 750 study nets per arm and measured by physical observation of the net in each room. All observed nets were recorded, and the householder was asked if the net was used for its intended purpose.

### 2.6. Fabric Integrity

Fabric integrity was defined as the physical state of the net to estimate bite protection. During surveys, the structured questionnaire was administered to each household and thereafter, each net was taken outside the room and hung in the frame by a trained technician. The nets were split into four different zones and holes were assessed using a hole template. The number and size of holes including tears in the netting and split seams by location and size were classified into four categories: smaller than a thumb (diameter of 0.5–2 cm, hole size 1), larger than a thumb but smaller than a fist (2–10 cm, hole size 2), larger than a fist but smaller than a head (10–25 cm, hole size 3), and larger than a head (>25 cm, hole size 4). Hole sizes greater than 0.5 cm were recorded [21]. The holes were counted from zone one (bottom part of the net), upwards to the roof section. All data were recorded in an Open data kit (ODK) version 1.26, San Diego, CA, USA), and thereafter, the net was returned to the room and the user was instructed to use the net until the next visit.

### 2.7. Data Analysis

All analyses were conducted using Stata version 18. Household characteristics were computed using proportional statistics. There were an additional 6 to 12 houses visited during the survey period that were not initially selected, and while these nets were included in the analysis of consent results, they were not considered in the assessment of functional survival.

Hole size was weighted to calculate the proportionate hole index (pHI) using the formula pHI = (1 × number of size 1 holes) + (23 × number of size 2 holes) + (196 × number of size 3 holes) + (576 × number of size 4 holes). The pHI was categorized based on recommended cut-off points into three categories [22] (Appendix A). The sum of the pHI in the good and damaged categories was presented as serviceable LLINs, while those in the “too torn” category were termed as unserviceable. Furthermore, the proportion of nets with at least one hole of any size was calculated per net brand per time point. The attrition rate was calculated as the proportion of study nets not present in the household during the survey period due to wear and tear and other reasons, divided by all study nets originally received, excluding nets lost to follow-up [19]. Reasons for net loss were also investigated [22]. For functional survival, nets present at each time point in serviceable conditions were considered, while survivorship was defined as nets present in the household during the survey period, regardless of the pHI category. Cox proportional regression models were fitted to predict the median functional survival and survivorship of each net and its hazard ratio. Functional survival was defined as a net still in serviceable condition, with a hole area <643 cm^2^, that was still in possession during the time of the survey. Survival time was calculated as the duration between the start of follow-up and when the event occurred (net loss) in years. For all physically inspected nets, the survey time was taken at the time of the event. If the net was not observed, the respondent was asked to estimate when the net was lost, disposed of, or given away.

### 2.8. Ethical Statement

This study was nested in a larger cRCT conducted in Misungwi. The cRCT received ethical approval from Kilimanjaro Christian Medical Collage, the National Institute for Medical Research (NIMR/HQ/R.8a/Vol.IX/2743), and the London School of Hygiene and Tropical Medicine (Ref: 16524). Informed consent to explain the purpose (objective) and nature of the study was read in Swahili and the local language if the household head did not understand Swahili. For those who consented, a signature or fingerprint was taken.

## 3. Results

### 3.1. Study LLIN and Household Enrollment

A total of 1154 households were enrolled for follow-up. Amongst these houses, 3072 study nets were labeled of which 767 were standard Interceptor, 772 Interceptor G2, 766 Olyset Plus, and 767 Royal Guard (see Figure 2, Table 1).

The total number of households selected one month post-distribution for net durability assessment was 1154. However, additional houses were visited (unintentionally) during each survey round: 12 houses at 6 and 24 months (totaling 1166 houses), 11 houses at 12 months (totaling 1165 houses), 6 houses at 30 months (totaling 1160 houses), and 10 houses at 36 months (totaling 1164 houses) (see Figure 2). The average number of people and sleeping places per household was similar across study arms, as was the population age distribution (see Table 1). More than half of the household heads had primary education, and this was consistent across study arms. House structures and characteristics were similar, with burnt brick walls, mud floors, and metal sheet roofs being the most common materials, while over 90% of income in all study arms came from fishing or farming (refer to Table 1). At each cross-sectional survey, consent was given in 938 (80%), 1071 (92%), 1039 (89%) 1160 (83%), and 882 (76%) households at 6, 12, 24, 30, and 36 months, respectively (see Figure 2). The remaining dwellings were either not found, vacant, householders refused, or interviewers asked to return later.

### 3.2. Attrition

During longitudinal surveys, all causes of net attrition rate and losses were assessed (Figure 3 and Appendix A). At six months, the majority of the nets lost were either given away to relatives (39% (95% CI: 23–58) for Interceptor; 33% (95% CI: 20–47) for Interceptor G2 and 15% (95% CI: 8–27) for Royal Guard) or used in another location (43% (95% CI: 26–61) for Interceptor; 26% (95% CI: 15–4) for Interceptor G2, and 42% (95% CI: 29–55) for Royal Guard) except for Olyset Plus, where most of the nets were discarded (69%, 95% CI: 59–77) at six months.

At twelve months, LLINs given away to relatives, used in another location, and used for other purposes were almost half of lost nets for Interceptor and Royal Guard, while for Interceptor G2 and Olyset Plus, the majority (66% of each net type) of nets were lost because they were discarded. From 24 to 36 months, discarding the net was the main reason for attrition with the highest (87% (95% CI: 84–89) and 90% (95% CI: 87–92)) for Olyset Plus and 74% (95% CI: 70–78) and 90% (95% CI: 87–92) for Royal Guard, respectively (see Figure 3, Appendix A).

Overall attrition rate (all-cause net loss) at 6 months post-distribution was lowest (6.3%, 95% CI: 5–9) for Interceptor nets compared to dual-AI LLINs (Interceptor G2 9.1% (95% CI: 7–12), Olyset Plus 17.9% (95% CI: 15–21), and Royal Guard 10.1% (95% CI: 8–13)). There was a drastic increase in attrition in Olyset Plus of which half of the nets were no longer present in the houses compared to Interceptor nets, which was not the case for Interceptor G2 and Royal Guard at 12 months. At the 24-month survey, 81.9% (95% CI: 79–85) of Olyset Plus and 60.1% (95% CI: 56–64) of Royal Guard were no longer present, compared to Interceptor nets. Overall attrition rates increased until 36 months with Olyset Plus being significantly worse (90.5%, 95% CI: 88–93; *p* < 0.001) compared to the standard Interceptor (Table 2).

### 3.3. Physical Integrity

At six months, over 90% of nets distributed were still in serviceable condition, except for Olyset Plus with 75% of nets discarded. These proportions decreased with time, with only 39% (95% CI: 35–44) of Olyset Plus in moderate or good condition at 12 months compared to 80% (95% CI 76–83) for control nets (Interceptor). Of the different dual-AI LLINs, Olyset Plus performed the poorest; 82% (95% CI: 74–88) were categorized as too torn 36 months post-distribution, compared to Interceptor nets at 52% (95% CI: 46–58), while 58% (95% CI: 51–63) of Interceptor G2 and 68% (95% CI: 61–74) of Royal Guard were too torn (Figure 4).

The proportion of nets with at least one hole increased from 6 months (Appendix A) to 24 months but no difference was observed in holes between 30 and 36 months. There was a significant difference in the proportion of standard Interceptor with at least one hole and Olyset Plus (OR: 1.5, 95% CI: 1.2–1.8, *p* < 0.001) at 6 months and 12 months (OR: 1.3, 95% CI: 1.1–1.6, *p* = 0.002). For Royal Guard, the proportion of nets with at least one hole was only significant at 6 months (OR: 0.7, 95% CI: 0.6–0.9, *p* = 0.010) compared to Interceptor.

### 3.4. Function Survival and Survivorship of the Assessed LLIN

The median functional survival for Interceptor, Interceptor G2, and Royal Guard was 1.9 years each, while for Olyset Plus, the median functional survival was 0.9 years (see Table 3). More than 80% of the study LLINs were still in the houses (survivorship) regardless of the size of holes after 6 months of use, and the proportion of survivorship decreased as net age with 37% survivorship for Interceptor G2, 18% for Royal Guard, and 10% for Olyset Plus compared to 37% for Interceptor nets after 36 months of use (see Appendix A).

After 3 years of net use, only 21.8% (95% CI: 19–25) of Interceptor nets were still in serviceable condition compared to 19.7% (95% CI: 16–23) for Interceptor G2, 3.9% (95% CI: 3–6) for Olyset Plus, and 8.6% (95% CI: 7–11) for Royal Guard (see Figure 5, Appendix A).

## 4. Discussion

This study provides a comprehensive evaluation of fabric integrity and survivorship of dual-AI LLINs in the Misungwi district, Tanzania. The reported net life spans (functional survival) from the present study fell below the WHO-recommended threshold of three years in operational settings, which was consistently observed across all four LLIN brands of net. Specifically, Interceptor, Interceptor G2, and Royal Guard each exhibited a life span of 1.9 years, while Olyset Plus showed the shortest life span with 0.9 years. The survivorship, considering all nets observed in households regardless of hole size, followed a similar trend, with Interceptor and Interceptor G2 at 2.4 years and Olyset Plus and Royal Guard at 1.9 years each. The main reason reported for the shorter functional survival was attrition due to the wear and tear of net fabrics. Half of Olyset Plus nets were in very poor condition (unserviceable) at 12 months while the other nets crossed this threshold after 30 months of follow-up. According to the WHO, unserviceable nets would provide little to no protection to the sleeper [23].

The overall functional survival of all LLINs evaluated in the present study was found to be less than three years. In the study area, we discovered additional mosquito nets still in their original packaging, obtained from various sources. In every household, there were new nets from a different brand than the ones we distributed. This might contribute to the swift discarding of nets, as residents had spare nets available to replace the ones provided in the study. Several other studies also reported shorter functional survival than that recommended by the WHO. For instance, more recently, a trial in Tanzania assessing several standard pyrethroid (Olyset, PermaNet 2.0et, and NetProtect) LLINs side by side across different districts reported a median functional survival of 2.0 for Olyset, 2.5 years for PermaNet 2.0et, and 2.6 years for NetProtect [24]. There was variation between LLINs in the rate of damage, lost bio-efficacy, and number discarded by households. Of all the nets assessed, Olyset nets were discarded in a higher proportion than PermaNet 2.0et and NetProtect as they were perceived to provide no protection when torn [24]. Another study in Zanzibar reported a median survival of 2.9 years in Unguja and 2.7 in Pemba of PermaNet 2.0 vs. Olyset nets [25]. Similarly, in Ethiopia, a median survival of 19 months was reported for standard LLINs [26]. In contrast, a study conducted in Nigeria reported higher functional survival rates in three areas surveyed (3.0 years in Nasarawa, 4.5 years in Cross River, and 4.7 years in Zamfara), and the difference between states was influenced by social–economic status and housing materials, rather than netting materials [27]. In the present study, the functional survival of Olyset Plus was by far the lowest, much shorter than what has been reported in any other studies and lower than a study conducted in different settings in Tanzania that evaluated the same brand of net [28]. For example, in Muleba, the functional survival was 1.6 years for Olyset Plus and 1.9 years for standard Interceptor nets. Multiple factors could account for the disparity between the two studies. First, in the current study, in the households we observed other new non-study nets, potentially leading to the replacement of LLINs with new, non-study nets, even within the damaged category. The evidence of using other LLINs has been documented in the main cRCT, of which 76.5% to 82.6% of other nets were being used at 24 months, which was not the case in a previous study. In the present study, the decrease in net survivorship over time for each net brand aligned with the reduction in net usage during the main RCTs, with Olyset Plus net usage being the lowest at 36 months (11%), while Interceptor usage was slightly higher (>30%) [29]. Secondly, differences in user behavior [21,28,30,31] may also explain those differences.

After three years of LLIN use, Olyset Plus, Royal Guard, and Interceptor G2 generally exhibited slightly higher attrition rates compared to the standard LLIN, Interceptor. The questionnaire assessing all causes of attrition highlighted that the majority of LLINs being discarded were due to wear and tear and this proportion was increasing over time. The other causes of loss, especially at the beginning of the follow-up, were used in other locations or given away to family or others. Finally, LLINs sold, stolen, and destroyed accidentally represented only a small proportion compared to other causes of net loss. A similar finding was reported in Ethiopia, where the attrition rate of a sub-sample was 48.8% after three years, with the reason being that the nets were too torn (physically damaged) for use, while 13% were used in other locations, and 12.8% were used for other activities [26]. The increased attrition rate due to the loss of fabric integrity has impacted malaria transmission in malaria-endemic regions in Kenya, where 40% of nets were extremely damaged after 12 months post-distribution [32]. In Tanzania, attrition was even higher, with fewer than 83% of bed nets distributed for daily use no longer present in households after 3 years, giving a median survival rate of 1.6 and 1.9 years for Olyset Plus and Olyset net, respectively [28]. This is comparable to a research initiative undertaken in Burkina Faso to evaluate permethrin-pyriproxyfen nets, where the study findings reveal that merely 13% of the distributed nets remain in households after a span of 36 months [33]. In Senegal, where Interceptor nets were lost mainly due to wear and tear [34], users reported that nets were disposed of as they believed they did not offer protection due to the accumulation of holes [34]. These findings contrast with the World Health Organization (WHO)’s former assumption of nets being present and functional for 3 to 5 years in the community [19].

The physical integrity of all distributed LLINs deteriorated with time, with 50 to 80% of the nets considered extremely torn after 36 months, according to brands. Olyset Plus were the most damaged nets followed by the dual AI LLIN, Royal Guard, Interceptor G2, and Interceptor, the pyrethroid LLIN. In contrast, longitudinal monitoring conducted in north-west Tanzania reported that 37% of Olyset net and 55% of Olyset Plus were considered extremely damaged (unserviceable according to WHO categories) [28]. A cross-sectional community survey conducted in Uganda reports that, after 25 months, there were no discernible differences in the physical durability when comparing long-lasting insecticidal nets with and without PBO (piperonyl butoxide) [35]. In all surveys, Interceptor G2 had a lower proportion of “too torn” nets compared to Olyset Plus. No significant differences in the proportion of holes were observed between Interceptor and Interceptor G2 at any timepoint. The results from a structured questionnaire administered during a survey in Zambia reported that the nets developed holes quickly due to the size (small nets compared to bed size) and material of the net [6]. The most significant explanatory factor for survival has been reported to be the combination of a better attitude to net care and exposure to messages related to nets [36]. The multifaceted evaluation provides valuable insights into the challenges and dynamics of LLIN durability, aiding in the ongoing efforts to optimize malaria prevention strategies.

## 5. Conclusions

The median functional survival for all classes of LLIN was less than two years, with Olyset Plus median survival of less than 1 year compared to the 3-year survival formerly assumed by WHO. The main reason for net loss was attrition due to wear and tear. Ranking the nets, Interceptor, the standard pyrethroid (reference net), and Interceptor G2 seem to display better physical integrity than the other two dual-AI nets. More development from manufacturers, oversight of quality, and donor investment are needed to enhance the textile durability of next-generation mosquito nets.

## Figures and Tables

**Figure 1 insects-15-00108-f001:**
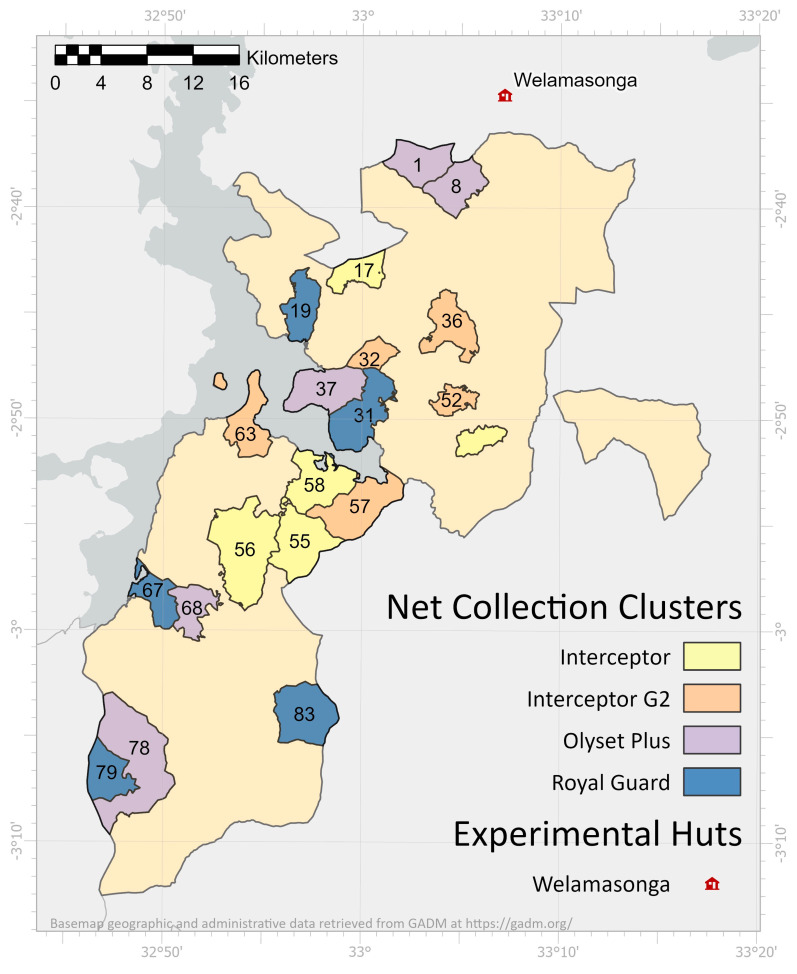
The 20 clusters randomly selected for net follow-up across Misungwi district: Olyset Plus (purple), Interceptor G2 (orange), Interceptor (yellow), and Royal Guard (blue). The numbers represent cluster numbers per arm where the nets were assessed. The map was created with ArcGIS software and all geographical and administrative data were sourced from GADM: https://gadm.org/license.html (accessed on 22 January 2024).

**Figure 2 insects-15-00108-f002:**
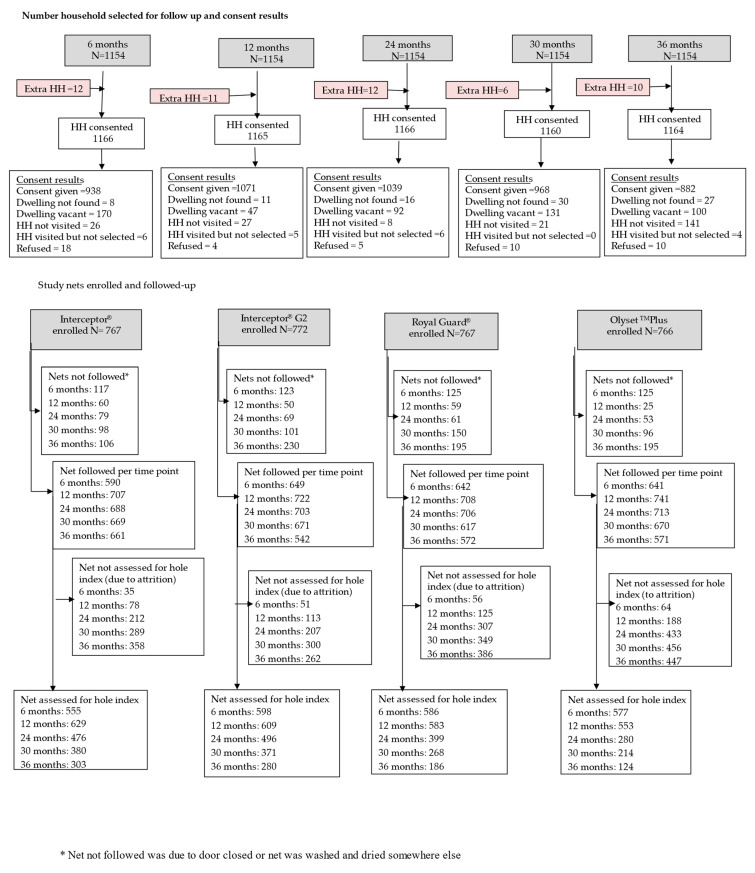
Number of households enrolled for follow-up and number of study nets assessed.

**Figure 3 insects-15-00108-f003:**
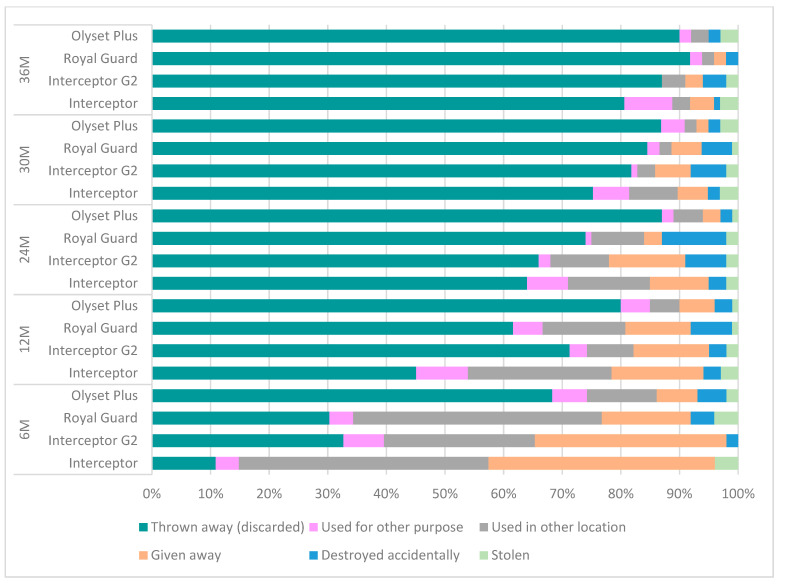
All causes of attrition by net type per survey.

**Figure 4 insects-15-00108-f004:**
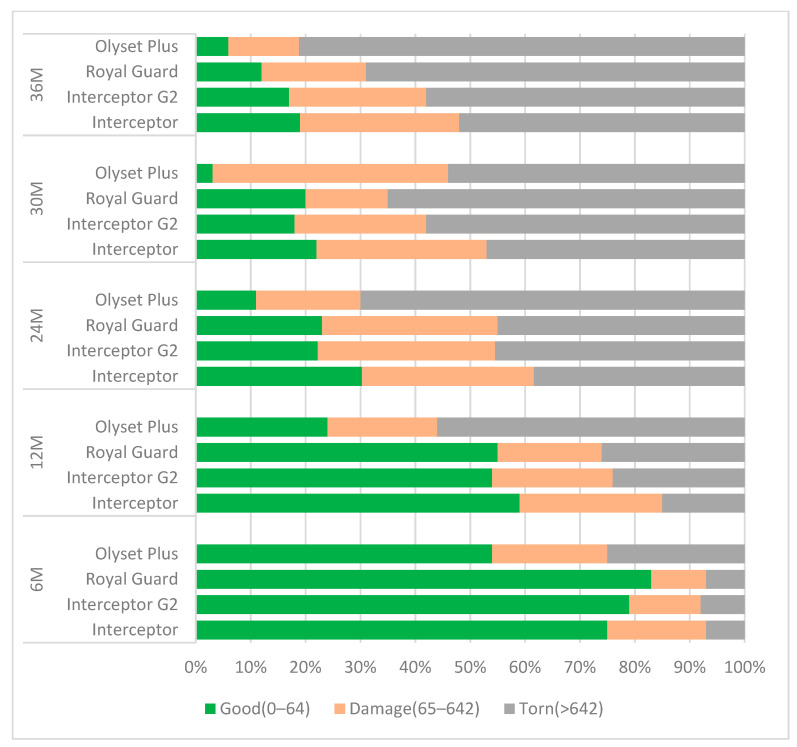
Physical condition of nets remaining in the household at the time of survey. Green shows proportion of nets in good condition (pHI 0–64), light pink shows proportion of nets in damaged condition (pHI 65–642), and grey shows proportion of nets in torn condition (pHI > 643). Nets in category “torn” are generally too torn to be useable, whereas nets in the categories good and damaged may still be used to good effect.

**Figure 5 insects-15-00108-f005:**
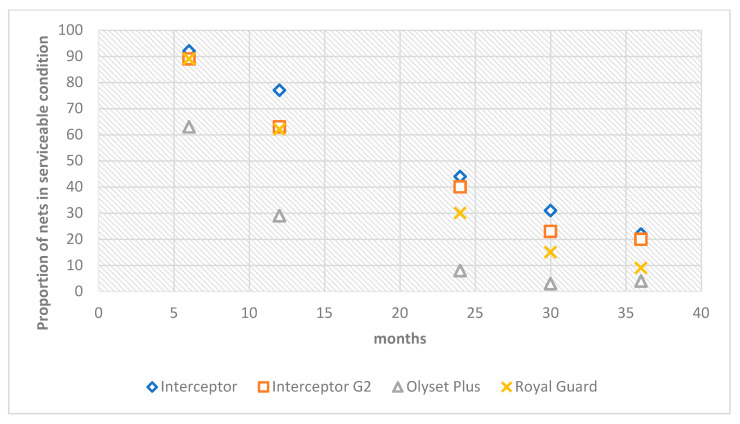
Estimated percentage of functionally surviving LLINs per time point.

**Table 1 insects-15-00108-t001:** Household, social, and economic characteristics of the study area.

Characteristics	Interceptor	Interceptor G2	Olyset Plus	Royal Guard
Number of participants	6624	6743	6604	6466
Average number of people per household	7.5	7.1	7.6	7.7
Mean number of sleeping places per household	3.7	3.5	3.5	3.6
Mean nets per household	2.76	2.61	2.71	2.57
Age distribution of household members % (95% CI)				
5 years	18.8% (17.9–19.5)	17.6% (16.8–18.5)	18.6% (17.5–19.7)	17.3% (16.4–18.2)
5–15 years	33.3% (32.3–34.4)	33.3% (32.1–34.6)	37.3% (34.6–40.1)	35.9% (34.6–37.2)
>15 years	47.9% (46.9–48.9)	49.0% (47.7–50.4)	44.1% (41.9–46.2)	46.8% (45.5–48.1)
Highest level of education of household head % (95% CI)				
No education	30.7% (27.6–34.1)	25.8% (22.8–29.0)	28.2% (25.1–31.5)	32.6% (29.5–36.1)
Primary education	66.6% (63.3–69.9)	69.3% (65.9–72.4)	69.7% (66.4–72.9)	64.6% (61.1–67.9)
Housing materials % (95% CI)				
Walls: burned brick	99.4% (98.5–99.7)	97.5% (96.2–98.4)	98.6% (97.5–99.2)	98.9% (97.9–99.5)
Floor: mud	61.2% (57.7–64.6)	62.4% (58.9–65.8)	72.0% (68.8–75.1)	69.9% (66.6–73.1)
Roof: metal sheet	76.9% (73.8–79.7)	70.7% (67.4–73.8)	72.4% (69.2–75.5)	72.4% (69.1–75.4)
Source of income % (95% CI)				
Fishing/farming	98.7% (97.6–99.3)	90.4% (88.2–92.3)	98.6% (97.5–99.2)	98.9% (97.9–99.5)

**Table 2 insects-15-00108-t002:** Percent attrition of LLINs surveyed and hazard ratio per net type and net age.

Net Type	% Attrition (95% CI)					Hazard Ratio
	6 Months	12 Months	24 Months	30 Months	36 Months	
Interceptor	6.3% (5–9)	15.9% (13–19)	40.6% (37–44)	52.8% (49–57)	62.9% (59–67)	1
Interceptor G2	9.1% (7–12)	21.1% (18–24)	43.2% (40–47)	57.9% (54–62)	63.3% (59–67)	1.4 (0.9–2.1), *p* = 0.121
Olyset Plus	17.9% (15–21)	50.7% (47–54)	81.9% (79–85)	85.2% (82–88)	90.5% (88–93)	2.8 (1.8–4.4), *p* < 0.001
Royal Guard	10.1% (8–13)	29.9% (27–33)	60.1% (56–64)	72.6% (69–76)	81.9% (79–85)	1.5 (0.9–2.4), *p* = 0.078

**Table 3 insects-15-00108-t003:** Median survivorship and functional survival of surveyed LLINs.

Net Type	Median Survivorship with 95% CI	Median Functional Survival with 95% CI
Interceptor	2.4 [2.4–2.7]	1.9 [1.9–2.0]
Interceptor G2	2.4 [2.4–2.5]	1.9 [1.9–1.9]
Olyset Plus	1.9 [1.8–1.9]	0.9 [0.9–1.0]
Royal Guard	1.9 [1.9–2.4]	1.9 [1.9–1.9]

## Data Availability

The data will be available in the LSHTM repository dataset. For now, there is ongoing secondary analysis until it is completed, then we will send the data to the LSHTM repository.

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
