# Peer review of "Monitoring of Fabric Integrity and Attrition Rate of Dual-Active Ingredient Long-Lasting Insecticidal Nets in Tanzania: A Prospective Cohort Study Nested in a Cluster Randomized Controlled Trial"

_insects, 2024, doi:10.3390/insects15020108_

Round 1
Reviewer 1 Report
Comments and Suggestions for Authors
The manuscript is very well written and brings interesting information about the durability of new bed nets with two insecticides incorporated into fibres. This is very important to the field since LLINs have been the main contributor to the reduction of malaria in Africa.
The authors have experience with the subject of the study and have published other papers. It also contributes to the organization of the information, clarity in describing the methods used and elaboration of tables and figures.
I only have one main question (about the results described in lines 255-262) and a few minor corrections and suggestions for the text.
Lines 255 – 262: if half of the nets were no longer present in the houses and this percentage increased in the following visits (reaching 90,5%) for all LLINs, how could the authors observe that high number of nets displayed in Figure 2 in each visit? I think the boxes entitled “net observed per survey” should display lower numbers at each time (12, 24, 30, 36 months) to be in accordance with what the authors said.
Figure 1: some numbers shown on the map are not legible.
Figure 2: some values are not correct. For Interceptor n=797, but if you sum up 590 (nets observed at 6 months) + 117 (nets not observed at 6 months) = 607. What happened to the 90 nets missing? For Royal Guard at 36 months n=767, but if you sum up 571 (nets observed at 36 months) + 195 (nets not observed at 36 months) = 766. I think the correct number is 572 nets observed, because if you sum up 386 + 186 = 572.
Table 1: what does “average household size” mean? The number of people living in the house?
Line 278: I believe the word “and” can be deleted.
Line 286: I suggest the authors write if the proportion of nets with at least one hole for Royal Guard was significantly higher or lower compared to Interceptor.
Line 316: there is an extra space – “survivorship ( all net”
Line 370: there is a lack of space – “Plus)and the”
Author Response
Please see the attached responses.

Reviewer 2 Report
Comments and Suggestions for Authors
The authors present an interesting study on the durability of four insecticide-treated bed nets by following the fabric integrity and attrition rate of the nets in a 36-month period in Tanzania. They found all of them have shorter life spans than 3 years and materials of net is the major driver. The sample size is very impressive, representing an excellent sampling effort. However, I found there are still some elements difficult to interpret in its current form.
Major issues:
The authors estimated the durability of nets in terms of fabric integrity and attrition rate. While they directly measured the fabric integrity as a hole index, the attrition rate was estimated through a questionnaire. Given that 20-30% of the households have no education background, how can the participant handle the questionnaires? If most the data collected from the questionnaires came from educated households, substantial biases can be caused and the contribution of social economic status to the durability of nets might have been underestimated. In addition, Given the authors did not measure what materials are in the bed nets and compare the durability of different materials, nor identify the relative contribution of materials of nets, social status and housing materials to the durability of nets in the same model, I feel their conclusion “Short life spans of all assessed LLINs were driven by material of the net, rather than social economic status and housing material” in the abstract is not directly supported by their findings.
Minor issues:
L20: three new bed nets (Interceptor® G2, Royal Guard® and OlysetTM Plus) with two insecticides incorporated into fibres …
L41: why pyrethroid only net (Interceptor ®) serves as the standard for the comparison to all other nets, given Interceptor® G2 does not contain pyrethroid.
L42: it is unclear why the total number of nets is as high as 40,000 here but only 3072 of them were followed later.
L48-49: I am not sure if their findings support this conclusion, especially they did not directly compare the durability of different net materials, nor directly identify the contribution of material of nets, social economic status and housing material in a statistical model.
L59: the in-text citation style looks unusual, it is either “(author, year)” or number in superscript without the parentheses. Please consider correcting it throughout the manuscript.
L85: “can penetrate and ingest blood…
L115, 118: 1/ 2/ looks weird
Study area: the rationale for study cluster selection is unclear. Consider explaining it more to clarify whether the selection is random and can represent sufficient replication for their experimental settings.
Figure 1: what are the clusters in dark color? Authors may consider removing the number and color for the clusters that are not involved in this manuscript.
L176: “0.5-2 cm” it is unclear whether this represents length or width of the hole, as the calculation of hole size involves both length and width, as mentioned in line 187.
Table 1: what is the unit for mean sleeping spaces per household
Table 1: when I calculate the total number of household by adding the estimated household numbers from four net types like this: (N of net type 1 from figure 2/ mean net type 1 per household from table 1)+ (N of net type 2/ mean net type 2 per household)+ (N of net type 3/ mean net type 3 per household)+ (N of net type 4/ mean net type 4 per household), the outcome is 910.8, why this number is much more lower than 1154?
L282: is the proportion of nets with at least one hole calculated by number of nets with holes/total number of nets within a household? Please specify it.
L283-287: these two sentences are confusing and what is OR? Consider rephrasing them.
Figure 5: can you show the survival curve for each net type and indicate whether there is significant difference between curves?
Discussion: in order to further improve the study, authors may consider discussing more about the relevance of their findings to previous studies and the significance of their study, e.g., thoroughly explain the disparities between studies and justify their findings?
how their findings may better inform decision making in malaria control?
Supplementary materials: many of them have incomplete information, e.g., months and confidence intervals not specified in appendix 2; no detailed information for the bar and standard errors/deviations in appendix 4.
Comments on the Quality of English LanguageGood
Reviewer 3 Report
Comments and Suggestions for Authors
Long-lasting insecticidal nets (LLINs) have been a major contributor to malaria reduction in sub-Saharan Africa over the past two decades. The development of pyrethroid insecticide resistance threatens the effectiveness of these LLINs, especially when holes develop in the nets and the insecticides degrade. This paper reports the physical durability of LLINs containing three classes of dual active ingredients compared to standard pyrethroid-only nets (Interceptor®). Approximately 40,000 nets of each type were distributed to various villages in Tanzania in February 2019, and a total of 3,072 LLINs were interviewed and net observed 6, 12, 24, 30, and 36 months later. Comparing the survival status of the net and the integrity of the fabric, it was found that the net containing dual active ingredients was physically vulnerable, and it was reported that the net's effectiveness was less than the three years recommended by the WHO.
The scale of the observations was large enough, the results were clear, and I think the data is important and worth publishing.
There are just a few points to be concerned about. I do not think this paper should be published unless these questions are dispelled.
Figure 2. Number of household and study nets enrolled for follow-up.
The biggest question is the numbers shown in Figure 2. It looks that the number of observed and unobserved bed nets is similar among the four groups. These numbers are not supposed to be linked between groups, so I feel this is intentional. When it comes to managing large amounts of data, especially data that can be manipulated arbitrarily, such as from questionnaires, I think it is necessary to take the time to randomly compare the raw data with the data used for analysis to check the soundness of the data. I think it is important to check what kind of consideration was or was not given to the soundness of the data in this study. Also, if consideration has not been given, I think the authors should check whether the data is correct.
Figure 3
I think it would be easier to see Figure 3 if it were shown as a bar graph like Figure 4. Not a pie chart. I also think that what the authors want to show is the proportion of discarded mosquito nets, so wouldn't it be better to use a noticeably different color? Isn't there a reason why there's no need to differentiate between items that have been given to someone or are being used somewhere else? Isn't it okay to use a similar color for similar reasons? If what we want to show here is that mosquito nets are physically damaged and are no longer being used or thrown away, they should create a graph that makes it easy to see that. Also, if ABCDE simply indicates 6, 12, 24, 30, and 36 months later, using ABCDE would actually make it more difficult to understand. Wouldn't it be better to just show the number of months?
Discussion
This argument is a bit too simplistic, and I think it is necessary to point out and discuss the contradictions with previous research reports. Cited References 15 to 18 are all published in well-known journals and evaluate the effectiveness of bed nets containing multiple active ingredients when used outdoors, compared to bed nets containing only pyrethroids. Some papers have concluded that it is also economically effective. These papers evaluated the effectiveness of bed nets for a shorter period of time than 36 months, but they all reported that they were effective for more than a year. Since this paper reports on the short lifespan of bed nets, it is necessary to consider why this is so, if it differs from previous data. Furthermore, the WHO recommends that bed nets last for 3 years, and rather than criticizing them for not meeting that requirement, the cited papers15-18 all did not follow up for 3 years, so the WHO recommends 3 years. I think we need to reconsider the year period. I think the material of the mosquito net will change along with the type of drug used. Regardless of whether bed nets containing multiple active ingredients are ineffective, the lack of discussion beyond reporting the physical weaknesses of bed nets that are reported to be effective gives the impression that something has been overlooked.
Round 2
Reviewer 2 Report
Comments and Suggestions for Authors
The authors have made thorough modifications to the manuscript, particularly in the discussion section, in response to the reviewers’ comments. I have only a few minor points this time.
1. In my previous comments, the concern regarding how the households that have no education background respond to the questionnaires and the potential biases caused by this limitation were a major issue. They have addressed this concern in their response, but this information should also be provided in the main text of the manuscript as clarification and critical information for readers.
2. L40: The authors answered that the Interceptor ® G2 net also contains pyrethroid, but I did not find this information in L40. Is the alpha-cypermethrin more specifically a pyrethroid alpha-cypermethrin? Please specify it.
3. In response to my comments, the authors provide the survival curves for the four types of nets in order to show if there is significant difference between them. I did not see any statistics within the figure to show if the curves have significant difference or not. This figure is valuable information and should be added to the supplementary documents as well.
4. Authors should explain whether the bars are standard errors or standard deviations in Appendix 4.
5. L346: the comma before “Multiple” should be a period.
Reviewer 3 Report
Comments and Suggestions for Authors
The data is important and worth to be published.
However, the authors did not reply to my previous question properly. I do not think this paper should be published unless these questions are clarified.
1. The biggest question is the numbers shown in Figure 2.
Because they are too similar. I do not write they are the same.
It is necessary to take time to randomly compare the raw data with the data used for analysis to check the soundness of the data.
In any kind of data handling, we would repeat the processes to check if there are any mistakes. I asked the authors if they did check and what kind of. But the authors only replied that the figures are not the same. Do the authors mean that they did not check the correctness of the data?
2. Figure 3 has been improved. Thank you.
3. Discussion
The discussion is greatly improved.
But this revision has raised some more questions.
>>
Olyset Plus and 1.9 years for standard Interceptor nets, Multiple factors could account for 346
the disparity between the two studies. First, manufacturing issues with LLINs and user 347
behavior may contribute. <<
Can the authors elaborate the meaning of manufacturing issues with LLINs and user behavior? Manufacturing with LLINs should be the same with other studies. Was user behavior different in the same country? Both reasons are hard to be believed. The authors must explain why they think in that way.
>>
Secondly, in the current study, we observed new packs of non- 348
study nets, potentially leading to the replacement of LLINs with new, non-study nets, 349
even within the damaged category. The evidence of using other LLIN has been docu- 350
mented in the main cRCT, of which 76.5% to 82.6% of other nets were being used at 24 351
months , which was not the case in previous study. In the present study, the decrease of 352
net survivorship over time for each net brand align with the reduction in net usage during 353
the main RCTs, with Olyset Plus net usage the lowest at 36 months (11% ), while intercep- 354
tor usage was slightly higher (>30%) (29).<<
This reason seems to be more important. Purchase of new LLIN is not involved in the reasons why villagers throw away the distributed LLINs. This part should be described in the methodology and in the results not in the discussion.
The point of this research is that the mosquito net usage period was much shorter in Tanzania (or in this survey).
The mosquito net using one insecticide also shows a short use period. It isn't essential to compare them because all mosquito nets have a very short period of use. None of them met the conditions recommended by W.H.O. The critical thing in this study can be to shed light on the short period of LLIN usage in the field.
Round 3
Reviewer 3 Report
Comments and Suggestions for Authors
Dear the authors,
Thank you for your detailed explanation to my questions.
I agree with the author's answer.
I consider the data reliable and believe this paper has high utility as it reports surprisingly short lifespans of bed net use at large spatial scales. The discussion has been greatly improved from the previous draft.
Since the producers of bed nets are putting in a lot of effort and trying to contribute to improving public health, researchers should also remember to respect their efforts. Disclosing the results objectively in a neutral manner will be a valuable opportunity for mosquito net manufacturing companies to learn about the results of their research and efforts. I hope that this paper will contribute to improving the quality of insecticide-soaked bed nets in future.
Minor points
L340 that , is not than?
L349 ,,(28).
L352 Firstin ,, is not First in ?
Author Response
Minor points
L340 that , is not than?
Response: Line 340, we change that to than (see below)
Of all the net assessed, Olyset net were discarded in high proportion than permanent and NetProtect as it was perceived to provide no protection when torn (24).
L349 ,,(28).
Response: Line 349, the dot and space is added after reference 28. (see below)
In the present study, the functional survival of Olyset Plus was by far the lowest, much shorter than what has been reported in any other studies and lower than a study conducted in a different settings in Tanzania which evaluated the same brand of net (28).
L352 Firstin ,, is not First in ?
Response: Line 352, we space the word firstin now read as first in.
First in the current study